# Computationally unmasking each fatty acyl C=C position in complex lipids by routine LC-MS/MS lipidomics

Leonida M. Lamp [1,8], Gosia M. Murawska[2,8], Joseph P. Argus [2,8], Aaron M. Armando [2], Radu A. Talmazan [3], Marlene Pühringer [4,5], Evelyn Rampler [4], Oswald Quehenberger[2], Edward A. Dennis [2,6,9] ✉ & Jürgen Hartler [1,2,7,9] ✉

Identifying carbon-carbon double bond (C=C) positions in complex lipids is essential for elucidating physiological and pathological processes. Currently, this is impossible in high-throughput analyses of native lipids without specialized instrumentation that compromises ion yields. Here, we demonstrate automated, chain-specific identification of C=C positions in complex lipids based on the retention time derived from routine reverse-phase chromatography tandem mass spectrometry (RPLC-MS/MS). We introduce LC=CL, a computational solution that utilizes a comprehensive database capturing the elution profile of more than 2400 complex lipid species identified in RAW264.7 macrophages, including 1145 newly reported compounds. Using machine learning, LC=CL provides precise and automated C=C position assignments, adaptable to any suitable chromatographic condition. To illustrate the power of LC=CL, we re-evaluated previously published data and discovered new C=C position-dependent specificity of cytosolic phospholipase $A_2$ (cPLA$_2$). Accordingly, C=C position information is now readily accessible for large-scale high-throughput studies with any MS/MS instrumentation and ion activation method.

Lipidomics, dedicated to the large-scale study of lipid metabolites, is rapidly gaining prominence in basic and translational research, consequently opening new avenues for prevention, prediction and treatment of various diseases[1–3]. Lipidomics is striving towards structural characterization in ever-increasing detail[4]. In particular, the localization of C=C positions in unsaturated complex lipids is a major focus of research, since mounting evidence demonstrates their critical role in various physiological and pathological conditions, including wound healing, inflammation, neurological disorders, cancer, as well as cardiovascular and metabolic diseases[5–13]. For example, phospholipase $A_2$ (PLA$_2$) specificity[14] in releasing specific fatty acids like arachidonic acid, leading to inflammatory mediators[15], depends on C=C positions. In fact, in early clinical trials, dietary omega-3 supplementation has been shown to be more effective when administered in the form of distinct complex lipid compounds rather than as unesterified fatty acids, for instance in reducing ADHD symptoms and premenstrual syndrome,

[1]Institute of Pharmaceutical Sciences, University of Graz, Graz, Austria. [2]Department of Pharmacology, University of California San Diego, La Jolla, CA, USA. [3]Laboratoire de Physique et Chimie Théoriques, Université de Lorraine, Nancy, France. [4]Department of Analytical Chemistry, Faculty of Chemistry, University of Vienna, Vienna, Austria. [5]Vienna Doctoral School in Chemistry (DoSChem), University of Vienna, Vienna, Austria. [6]Department of Chemistry and Biochemistry, University of California San Diego, La Jolla, CA, USA. [7]Field of Excellence BioHealth, University of Graz, Graz, Austria. [8]These authors contributed equally: Leonida M. Lamp, Gosia M. Murawska, Joseph P. Argus. [9]These authors jointly supervised this work: Edward A. Dennis, Jürgen Hartler. ✉e-mail: edennis@ucsd.edu; juergen.hartler@uni-graz.at

normalizing blood lipids, and lowering inflammation markers like C-reactive protein[16,17]. However, with routine lipidomics, this essential structural resolution between omega-3 and omega-6 fatty acyls in complex lipids is still not achievable.

Similar to quantitative proteomics, comprehensive profiling of thousands of unique lipids must cover concentration ranges of six to eight orders of magnitude[18,19]. A widely used approach that meets this required sensitivity in a high-throughput setting couples reverse-phase liquid chromatography with tandem mass spectrometry (RPLC-MS/MS). This setup is capable of discriminating between thousands of isobaric and isomeric lipids, which are common in lipidomics analyses of complex biological samples. Specifically, lipids can be composed of a combination of a variety of building blocks, including lipid backbone, fatty acyl, alkyl or alkenyl chains (FA), and polar head groups. Utilizing RPLC-MS/MS, sufficient structural information can be resolved to report lipids at the FA level (e.g., PC 16:0_20:4), and/or FA position level (e.g., PC 16:0/20:4 or PC 20:4/16:0), collectively termed as identifications at the *lipid molecular species* level[20].

The physicochemical properties of complex lipid species are strongly affected by the position of C=C in unsaturated FAs[21], but they cannot be determined presently by routine MS/MS. To determine C=C positions in complex lipids, there are currently two principal strategies: gas-phase ion activation/dissociation and chemical derivatization of C=C bonds. The former include electron-activated dissociation (EAD)[22], oxygen attachment dissociation (OAD)[23], ozone-induced dissociation (OzID)[24], radical-directed dissociation (RDD)[25] and ultraviolet photodissociation (UVPD)[26]. Typical examples of the latter are based on the Paternò–Büchi (PB)[27] and epoxidation[28] reactions. While these strategies are able to identify C=C positions in complex lipids, routine high-throughput implementations of these approaches have been impeded for multiple reasons: (1) all of these methods require specialized instrumentation or chemistry[22–28]; (2) several involve the use of hazardous chemicals[24,28]; (3) some only work in positive ion mode[22], while several lipid classes ionize better in negative ion mode; (4) data analysis is complicated by the inherently increased complexity at the MS[1] and/or MS[n] level[22–28]; (5) unambiguous assignment of the C=C position to the respective FA is frequently not feasible when more than one unsaturated chain contributes to a spectrum[29]; (6) yield of

characteristic ions is low due to either limited dissociation or derivatization efficiencies[22–28]; (7) measured quantities are in many cases not comparable[29,30]; (8) sensitivity is reduced in comparison to conventional methods[22–28].

In this study, we present a universal solution to the limitations described above by introducing the "LDA C=C Localizer" (LC=CL, Fig. 1), an automated tool for FA-specific identification of terminal C=C locations (ω-positions) based on retention time (RT) information from routine RPLC-MS/MS data. The LC=CL extension is designed as an integral part of the open-source Lipid Data Analyzer (LDA) software[31,32], which has been embraced by the lipidomics community as a prime tool for automated lipid identifications[33]. Through LDA's flexible decision rule concept for confident identification of lipids[32], the LC=CL is applicable to any RPLC-MS/MS setup, and also, as recently demonstrated, to any ion activation method[34]. Accompanying LC=CL, we provide the most comprehensive database of experimentally derived ω-position resolved lipid species, encompassing more than 2400 complex lipid species, of which 1145 are novel identifications. Leveraging the RT profile captured in this database, LC=CL facilitated the discovery of previously undetected C=C position specificity of cytosolic phospholipase A$_2$ (cPLA$_2$)[21], a critical player in the production of bioactive lipid mediators.

## Results

### The LC=CL workflow

Automated FA-specific ω-position resolved lipid identification by RT information is realized via the three key software components illustrated in Fig. 2a. (1) Experimentally verified RT databases (RT-DB) serve as a reference for an ω-position specific elution profile of complex lipids. LC=CL features a module for the automated creation or extension of our comprehensive reference RT-DB. (2) The developed machine learning algorithm guarantees the accuracy of the RTs of the individual ω-position resolved complex lipids under various chromatographic conditions (Methods and Supplementary Note 1). Similar to the widely used DIA-NN proteomics software for identification of peptides from data-independent MS/MS acquisitions[35], this algorithm maps RTs to the new elution profile based on observed "anchor species". (3) The automated ω-localization is integrated in LDA's lipid

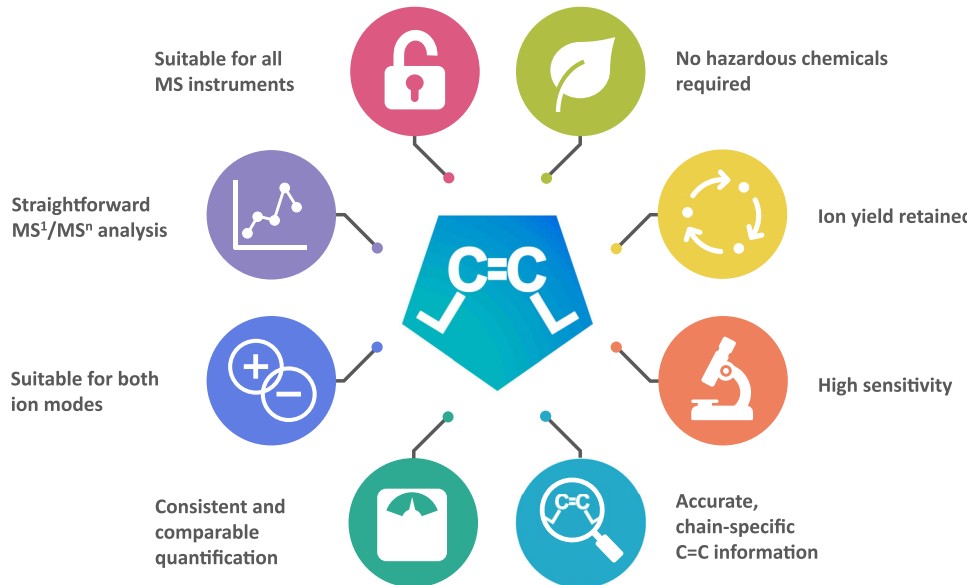

**Fig. 1 | Advantages of LC=CL's innovative ω-position identification strategy.** The LC=CL comes with no conceivable limitations in experimental MS/MS setups, as it is suitable for all MS instruments capable of MS/MS, irrespective of the MS analyzer or ion activation method used. LC=CL's chain-specific C=C position identification strategy retains the full sensitivity of the corresponding MS method, is suitable for both positive and negative ion modes and requires no hazardous chemicals. Additionally, MS[1] and MS[n] analysis is unaffected by the proposed method, translating to consistent and comparable quantification.

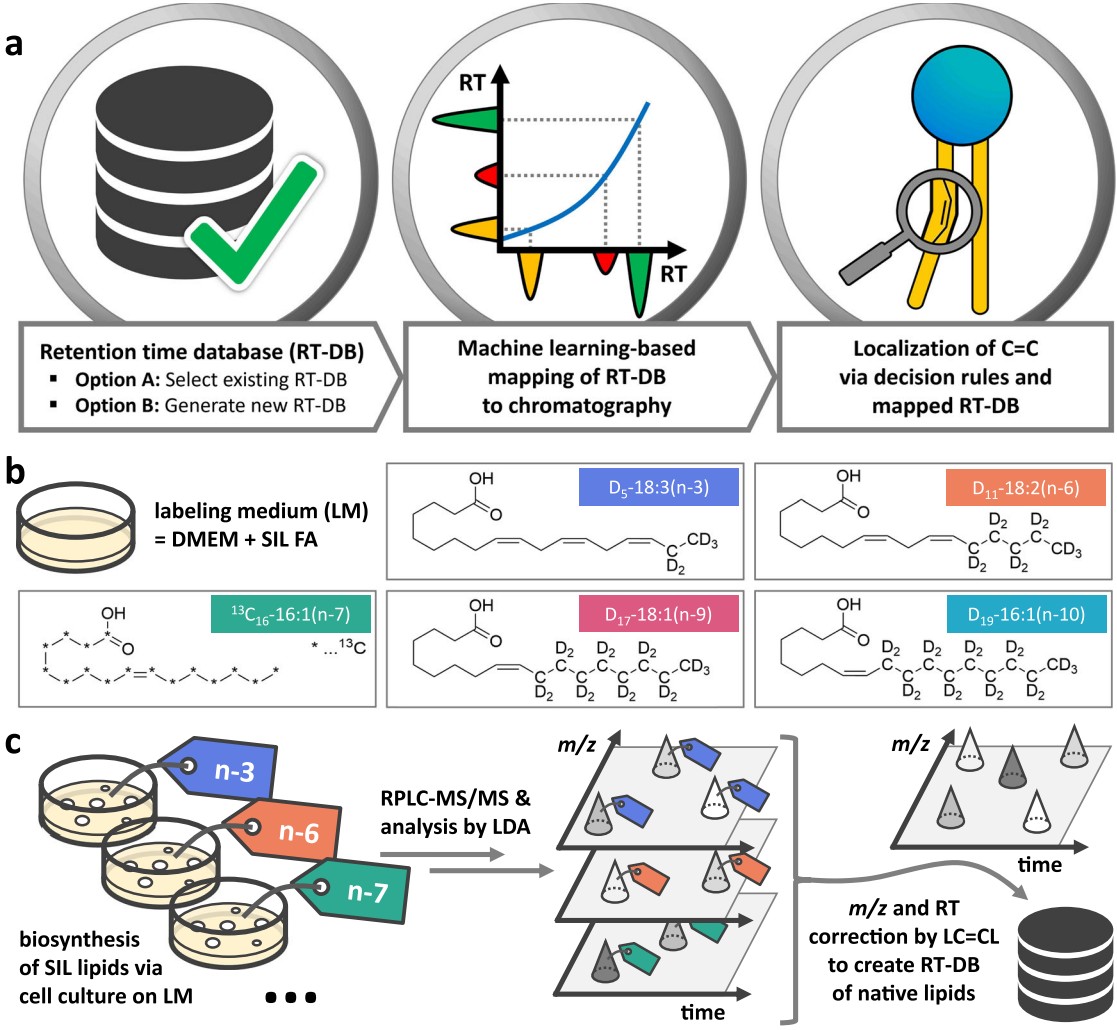

**Fig. 2 | The LC=CL analysis workflow includes RT-DB generation by SIL experiments.** Overview of the LC=CL analysis pipeline with a detailed scheme of the analytical workflow that was used to generate the RT-DB provided in this study. **a** The LC=CL workflow. To automatically localize lipid C=C in raw LC-MS/MS data, first, an RT-DB must be chosen (or newly generated). Next, the selected RT-DB is mapped to the relevant chromatographic conditions. In the final step, LC=CL fully automatedly localizes lipid ω-positions using LDA's decision rules and the mapped RT-DB. **b** The workflow for generating a new RT-DB with LC=CL's designated module starts with the preparation of labeling medium (LM) containing SIL FA precursors. In this study, SIL α-linolenic acid $D_5$-18:3(n−3), linoleic acid $D_{11}$-18:2(n−6), palmitoleic acid $^{13}C_{16}$-16:1(n−7), oleic acid $D_{17}$-18:1(n−9), and sapienic acid $D_{19}$-16:1 (n−10) were used. **c** Cell cultures (here RAW264.7 cells) are grown on LM for the biosynthesis of complex SIL lipids. RPLC-MS/MS and lipid annotation by LDA reveal the labels, which unambiguously indicate the ω-position. LC=CL corrects for the *m/z* shift and isotope effect on RT introduced by the SIL in silico to create an RT-DB of native lipids with known ω-positions.

identification and quantification routine[31,32]. In this step, RT mappings of (2) are utilized to assign ω-positions for lipid molecular species confirmed by LDA's decision rules (based on $MS^n$ evidence).

**Creation of ω-position resolved retention time database (RT-DB)**
To establish the experimentally verified RT-DB, we leveraged stable isotope-labeled (SIL) lipids produced by cells to address the lack of commercially available authentic standards for complex lipids with defined ω-positions. To this end, we supplemented RAW264.7 cells with SIL FAs followed by RPLC-MS/MS analysis and data processing with LC=CL (Fig. 2b, c and Methods). The cells elongate and desaturate the labeled FAs, while retaining the ω-position, and incorporate them into complex lipids. RAW264.7 cells are particularly well-suited for this purpose, as they express a wide variety of FA desaturation enzymes[36–38]. We used five commercially available SIL FAs (Fig. 2b), with deuterium labels located at the ω-end (methyl-terminus) of four of them and $^{13}C$ in all of the carbons of the 16:1 n-7 SIL-FA. This label positioning preserves the isotope tag during metabolic processing, as FA elongation, further desaturation, and the major truncation reaction

(β-oxidation) exclusively occur at the carboxy-terminus in mammalian cells[39]. The n−3, n−6, n−7, n−9, and n−10 SIL-FAs introduce a mass shift in MS analysis of five, eleven, sixteen, seventeen, and nineteen Da, respectively. After 24 h, the lipids were extracted and measured by RPLC-MS/MS[14].

The mass shift introduced by the label, which is reflected in the precursor mass and the fragments in the MS/MS spectrum, unambiguously indicates the ω-position. From the identified ω-position resolved complex lipids (Supplementary Data 1), RT data were collected and integrated into an RT-DB by LDA's LC=CL module, which corrects the isotope effect on retention time in silico (Fig. 2c and Methods). Notably, we applied two SIL-FA supplementation strategies: (i) To ensure comprehensive coverage of available lipid species, cells had to utilize substrates less favored under conventional dietary conditions to increase the production of low-abundant lipid species. Therefore, we supplemented each cell culture with a single type of SIL FA. However, this comes with the limitation that combinations of FAs with different ω-positions in a complex lipid (e.g., PI $D_{17}$-18:1(n−9)/$D_{11}$-20:4(n−6)) were not directly detected in a single experiment. The LC=CL automatically

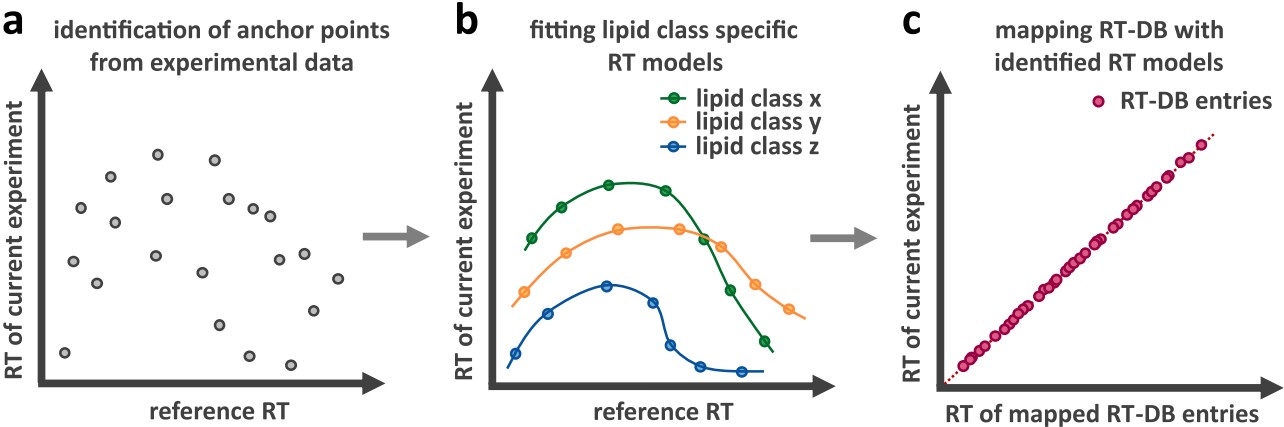

**Fig. 3 | RT-DB mapping procedure. a** In step 1, LC=CL identifies anchor species in experimental data (measurements of standards or biological samples). **b** In step 2, the algorithm fits RT models tailored to individual lipid classes. **c** These models are then used in step 3 to map the RT-DB to the chromatography of the current experiment.

infers such ω-position combinations by computationally aggregating the results of the individual SIL-FA experiments. (ii) To validate the computationally derived FA combinations with differing ω-positions, we also conducted experiments with SIL FAs supplemented in pairs: 18:3(n−3) and 18:2(n−6); 18:2(n−6) and 16:1(n−7); 16:1(n−7) and 18:1 (n−9); 16:1(n−7), and 16:1(n−10). The results of these experiments confirmed the computationally predicted ω-position combinations (Supplementary Fig. 1). A repeat of these supplementation experiments with native FAs allowed for the fully automated identification of more than 2000 ω-resolved lipid species. Relative quantitative increases of each ω-series reflected the supplemented FA-precursor, confirming that correct assignments were made (Supplementary Figs. 2 and 3).

### Machine learning-based RT-DB mapping

The LC=CL maps the experimentally obtained RTs stored in the RT-DB to the elution profile of the respective experiment. To this end, the algorithm identifies lipid species in experimental data of standards or biological samples that are suitable to serve as anchor points (Fig. 3 and Methods). Based on these anchor species, the elution profile mapping is performed via machine learning using a cubic spline interpolation model (Fig. 3 and Methods), which adapts flexibly to various chromatographic conditions and supports RT mapping tailored to individual lipid classes (Supplementary Note 1). Notably, interpolation with cubic splines has been shown to outperform alternative machine-learning approaches, such as deep neural networks (DNN) and multivariate adaptive regression splines (MARS) by a considerable margin when noise levels in the data are low, independent of data size, since they are efficient even when data is sparse[40]. The identified RT models are then applied to map the elution profile of the RT-DB to the current chromatography (Fig. 3). Importantly, to ensure high reliability of annotations, we abstained from predicting RTs of species that have not been experimentally verified, i.e., the elution profile of all possible LC=CL identifications have been verified by orthogonal means, such as the SIL experiments.

To evaluate the performance of the machine learning algorithm in calibrating an RT-DB to varying chromatographic conditions, experimental data were acquired using different batches of mobile and stationary phases, as well as 30- and 60-min chromatographic gradients. The calibrated RTs consistently deviated by only 1 s (median) and 2 s (average) from the true value within the same gradient, and by 3 s (median) and 5 s (average) across differing chromatographic methods. In comparison to that, ω-position isomers were typically separated by 10 s or more (Supplementary Fig. 4). These results (Supplementary Note 1) demonstrate that our machine learning algorithm can reliably calibrate our experimentally verified RT-DB toward differing experimental setups, thus making ω-position information readily accessible.

### Results of stable isotope labeling experiments

We identified 66 distinct unsaturated even-chain FAs in the phospholipidome of RAW264.7 cells (Fig. 4a). Forty-six of them fully align with the complete set of even-chain n−3, n−6, n−7, n−9, and n−10 FAs previously reported in RAW264.7 cells using a derivatization-based approach[38,41]. Notably, our findings also unveil 20 FAs that, to the best of our knowledge, were not yet documented in RAW264.7 cells (highlighted by a star in Fig. 4a, spectra provided in Supplementary Fig. 5).

Using this approach, we identified a total of 2408 complex phospholipid species with experimentally verified chain-specific ω-positions (Fig. 4b and Supplementary Data 1). Remarkably, 1145 represent novel lipid species with experimentally confirmed ω-locations that were not reported elsewhere, neither in the LMSD[42], RefMet[43], or HMDB[44] databases for RAW264.7 cells or any other organism (Fig. 4c and Supplementary Data 2). Of note, lipid analysis with LC=CL is not limited to the ω-position resolved species in the current RT-DB. The LDA environment supports the identification of virtually any additional lipid class and any possible FA combination[32,45].

### Benchmarking of LC=CL with EAD, PB, and OzID on human plasma

To demonstrate the transferability of our comprehensive RT-DB to other organisms and tissues, we profiled the ω-position-resolved phospholipidome of NIST SRM 1950 human plasma. We used these data to benchmark our approach versus three established C=C localization methods, EAD, PB, and OzID (Fig. 5). Our global lipid profiling approach[14] used with LC=CL provided a complete picture of the phospholipidome. Similarly, analysis with EAD was performed using a global profiling approach as described in the methods section. For the two PB studies, experimental conditions tailored to the lipid classes under investigation were used: The first PB study[27] focused on phosphatidylcholine (PC) and phosphatidylethanolamine (PE), including their ether variants, whereas the second study[46] exclusively reported on phosphatidylinositol (PI) species in human plasma. The OzID study[47] covered PC, PE, and PI.

The LC=CL identified a significantly higher number of species for the benchmarked classes PC, PE, and PI in a single study (Fig. 5a). For the species found in common, the ω-identifications were in excellent agreement across the orthogonal methods, with EAD results coinciding with all of LC=CL's ω-position assignments (see Methods and Supplementary Data 3 for a detailed list of identifications).

While the 30-min chromatography used in this study separates ω-positional isomers such as PC 18:0/22:5(n−3) and PC 18:0/22:5(n−6) by more than a minute (Fig. 5b and Supplementary Fig. 6), coelutions with other lipid molecular species are frequent, as is typical in lipidomics

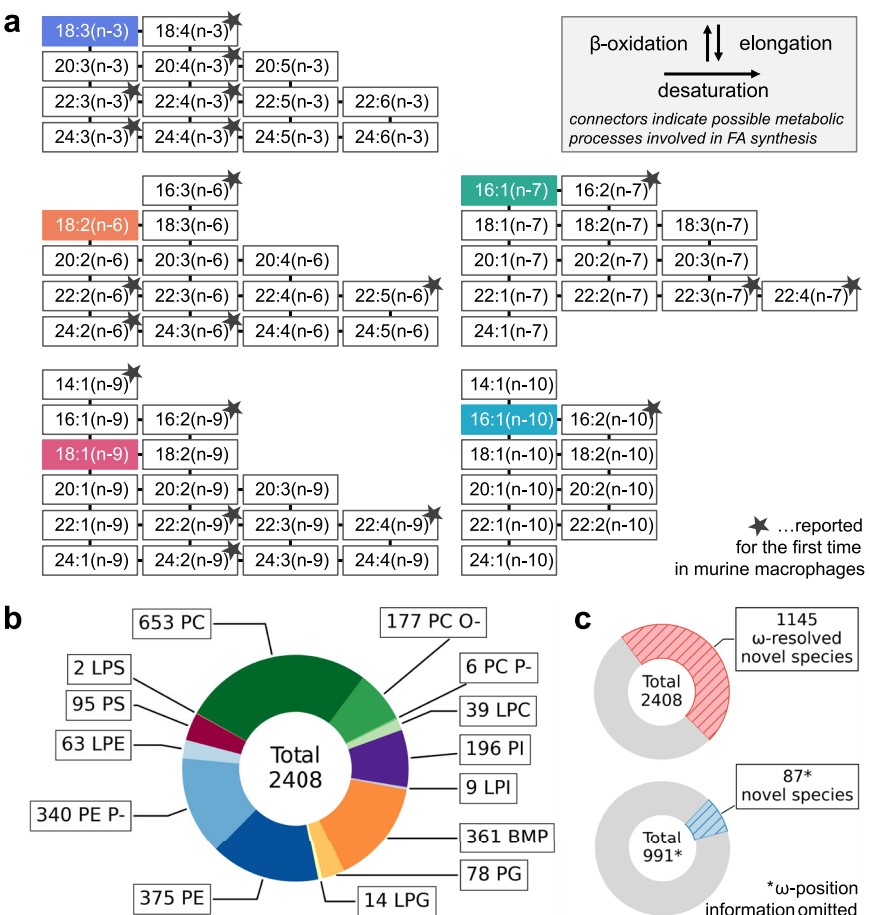

**Fig. 4 | Overview of detected species with experimentally verified ω-positions available in LC=CL's RT-DB. a** The supplemented SIL FAs served as the metabolic precursors for 66 distinct even-chain FAs, produced by the enzymatic machinery in RAW264.7 cells, which includes β-oxidation, elongation, and desaturation. Highlighted in color are the respective supplemented FAs. Stars indicate FAs that were not previously reported in RAW264.7 cells[38,41]. The listed FAs were detected in phospholipid species that are included in the RT-DB. **b** The comprehensive RT-DB contains a total of 2408 entries with experimentally verified ω-positions, encompassing the following 14 phospholipid subclasses: PC, PC O-, PC P-, LPC, PI, LPI, BMP, PG, LPG, PE, PE P-, LPE, PS, LPS. **c** Out of the total number, 1145 are novel complex lipid species that were not yet reported for any organism in either LMSD, RefMet, or HMDB databases (hashed red). When omitting ω-position information (hashed blue), 87 novel complex lipid molecular species were discovered.

---

analysis. In the provided example, PC 18:0/22:5(n−3) coelutes with PC 16:0/24:5(n−6). Such coeluting isomers are readily identified by the LC=CL with high confidence, as the algorithm assigns ω-positions only when MS/MS evidence is available for lipid molecular species. As this example illustrates, knowledge of the sum composition alone, i.e., PC 40:5, does not suffice for an assignment of C=C positions based on RT. In contrast, while C=C position identification of the major species, PC 18:0/22:5(n−3), is also quite confidently possible with EAD, this is not the case for the minor species (Fig. 5c). In cases of two similarly abundant coeluting species where both ω-positions can be identified, it is generally impossible to assign the C=C positions to either of the two isomers based on MS²-EAD fragmentation alone, as the indicative fragments can originate from either species. The same is true for lipid molecules with more than one unsaturated chain.

**Elucidating C=C positional specificity of cPLA₂**
To confirm the general applicability of our method, we re-analyzed data of a previously published study that investigated PLA₂ substrate specificity in RAW264.7 cells, reporting that (Group IV A) cPLA₂ targets arachidonic acid (AA) at the *sn*-2 position[14]. In this study, experimental data were acquired using a 30-min chromatography.

Using LC=CL, we were able to pinpoint the structural features that dictate cPLA₂'s substrate preference in detail. First, we verified that the chain length, saturation, and the position of C=C in the FA at the *sn*−1 position do not significantly influence cPLA₂ activity toward the FA at *sn*−2 (Supplementary Fig. 7a, b). Thus, we focused solely on the FA at the *sn*−2 position in subsequent analyses.

We discovered that cPLA₂ exhibited a similar specificity for mead acid (MA, 20:3(n−9)) as it does for its well-known substrate AA (Fig. 6a, b), a finding that has not been previously documented. Furthermore, upon cPLA₂ activation with KDO₂ Lipid A (KLA), we observed downstream elongation and desaturation products of AA and MA (Supplementary Fig. 7c, d). This observation further corroborates the finding that both AA and MA are released by cPLA₂. In contrast, the C=C positional isomers of MA, 20:3(n−7) and 20:3(n−6) (Fig. 6; spectra provided in Supplementary Fig. 8), are poor substrates for cPLA₂. In a previous study, our molecular dynamics simulations identified the first double bond at Δ5 in AA to be crucial for cPLA₂ activity[21]. This hypothesis is in line with our findings, as for both compounds located at the *sn*-2 position, AA and MA, the first double bond at the *sn*-2 alpha-carbonyl ester linkage is at the Δ5 position. In contrast, for the other two 20:3 C=C positional isomers, which are not cPLA₂ substrates, the first double bond for homo-allylic FAs is at position Δ7 or Δ8 (Fig. 6b). Accordingly, the observed cPLA₂ specificity for MA is sound from a mechanistic perspective. The ω-positions of PI 18:0/20:3(n−6), PI 18:0/20:3(n−7), and PI 18:0/20:3(n−9) were confirmed by

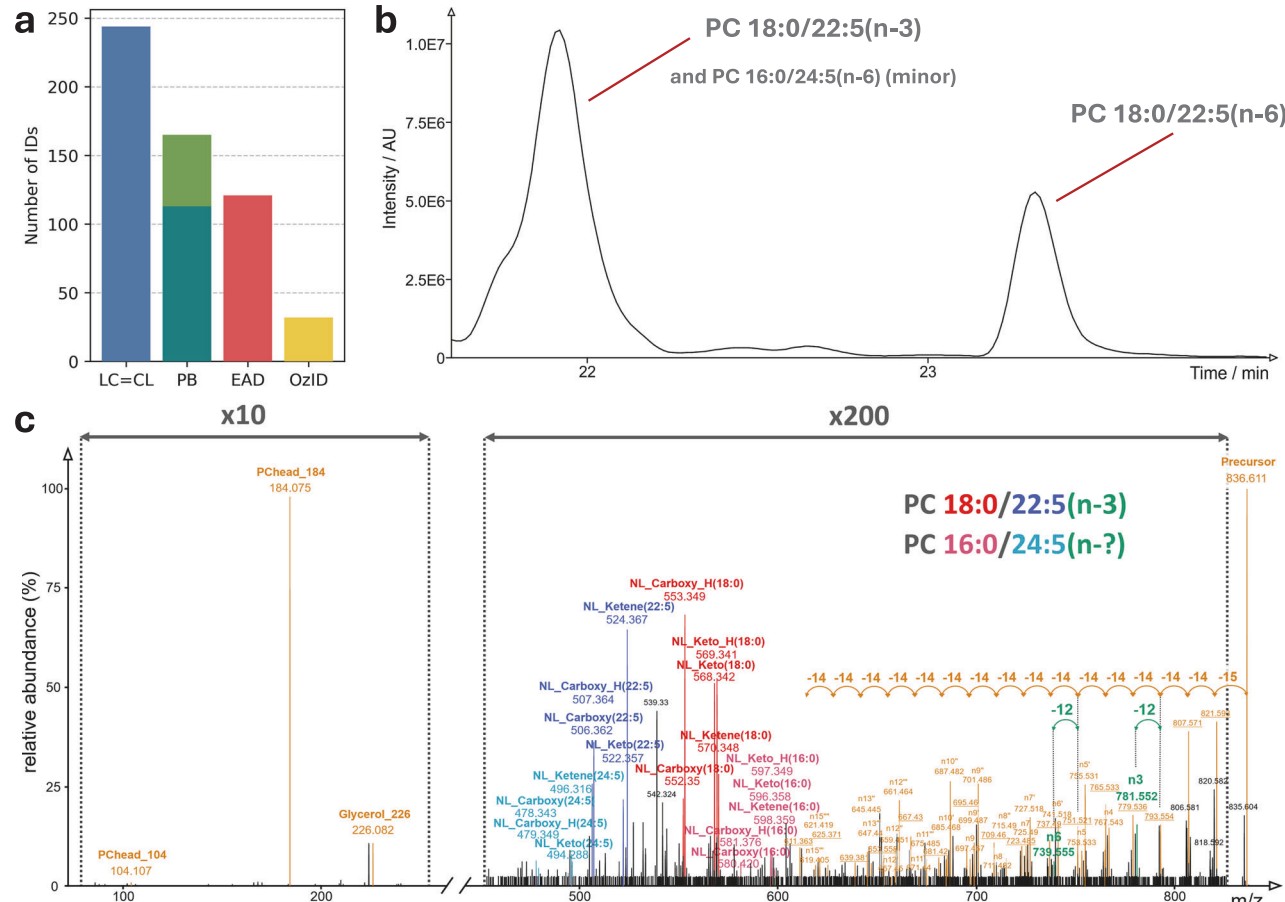

**Fig. 5 | Benchmark of LC=CL versus EAD, PB, and OzID. a** Total number of ω-resolved PC and PE, including their ether and vinyl ether variants, and PI identified in human plasma. Two separate studies contributed to the total number of identifications with PB: The first PB study[27], indicated in teal color, was tailored to phosphatidylcholine (PC) and phosphatidylethanolamine (PE), including their ether variants. The second study[46], indicated in green color, reported exclusively on phosphatidylinositol (PI) species in human plasma. **b** Chromatographic separation of ω-positional isomers in RAW264.7 cells using the 30-min chromatography at the example of PC 18:0/22:5(n−3) and PC 18:0/22:5(n−6). With PC 18:0/22:5(n−3), a minor species coelutes, which was identified as PC 16:0/24:5(n−6) by LC=CL

(chromatogram taken from the file '250213_RAW_n_6_30min_IDA_EAD_pos_01'). **c** EAD spectrum acquired at 21.9 min, precursor $m/z = 836.610$ (same file origin as in subfigure **b**) shows the fragments indicative of the two coeluting PC isomers. While the ω-position of the major species PC 18:0/22:5 can be unambiguously identified as n−3, the indicative fragment for the ω-position of the minor species is close to noise. The fragments depicted in red are indicative of FA 18:0, in blue for FA 22:5, in rose for FA 16:0, and in cyan for FA 24:5. Highlighted in teal green are the fragments 'n3' and 'n6', which are indicative of C=C at positions n−3 and n−6, respectively. Depicted in orange are other fragments resulting from the fragmentation of FA chains and fragments specific for the lipid class, PC.

supplementation experiments with native FA precursors (Supplementary Fig. 3). Moreover, all C=C positions as illustrated in Fig. 6b were confirmed with a targeted MRMHR approach using EAD (Supplementary Fig. 9 and Methods). Furthermore, this finding suggests that by employing LC=CL, the detailed specificity of all phospholipases and related enzymes can now be elucidated at a greater level of structural detail than previously imagined. Additionally, these results also demonstrate the universality and impact of the computational LC=CL approach in the analyses of routine mass spectrometry-based lipidomics data.

## Discussion

To understand biological processes in both physiology and pathophysiology, knowledge of C=C positions in lipids containing unsaturated FAs at the system level is indispensable. However, until now, information about ω-positions could only be derived by using specialized instrumentation or derivatization, both of which compromise sensitivity and often quantifiability, limiting their use in comprehensive lipidomics analyses. In this study, we demonstrate that the FA-specific positions of C=C are inherently encoded in the elution profile of complex lipids in routine RPLC poised for C=C position assignment.

We introduce LC=CL, a software solution that unmasks FA-specific ω-position information embedded in routine RPLC-MS/MS data, translating it into lipid structures. LC=CL's computational approach offers platform independence and utmost flexibility by circumventing the need for specialized instrumentation or derivatization. LC=CL's machine learning algorithm (Methods and Supplementary Note 1) guarantees adaptability to any chromatographic conditions, with small deviations of only a few seconds from the true RT-value. Thus, LC=CL makes ω-position information accessible to a variety of chromatographic conditions.

The RT-DB underlying LC=CL captures the elution profile of over 2400 lipid species, forming the most comprehensive collection of experimentally verified FA-specific ω-resolved complex lipid species. This RT-DB contains more than 1100 compounds that were not previously reported. This high number of novel ω-resolved identifications can be attributed to two factors: (i) the uncompromised sensitivity inherent with our approach; (ii) FA supplementation increases the synthesis of typically low-abundant species, and as such, the probability of detecting them. Of note, also two of the lipid species identified by LC=CL in the previously published dataset, which were instrumental in corroborating the C=C-dependent substrate

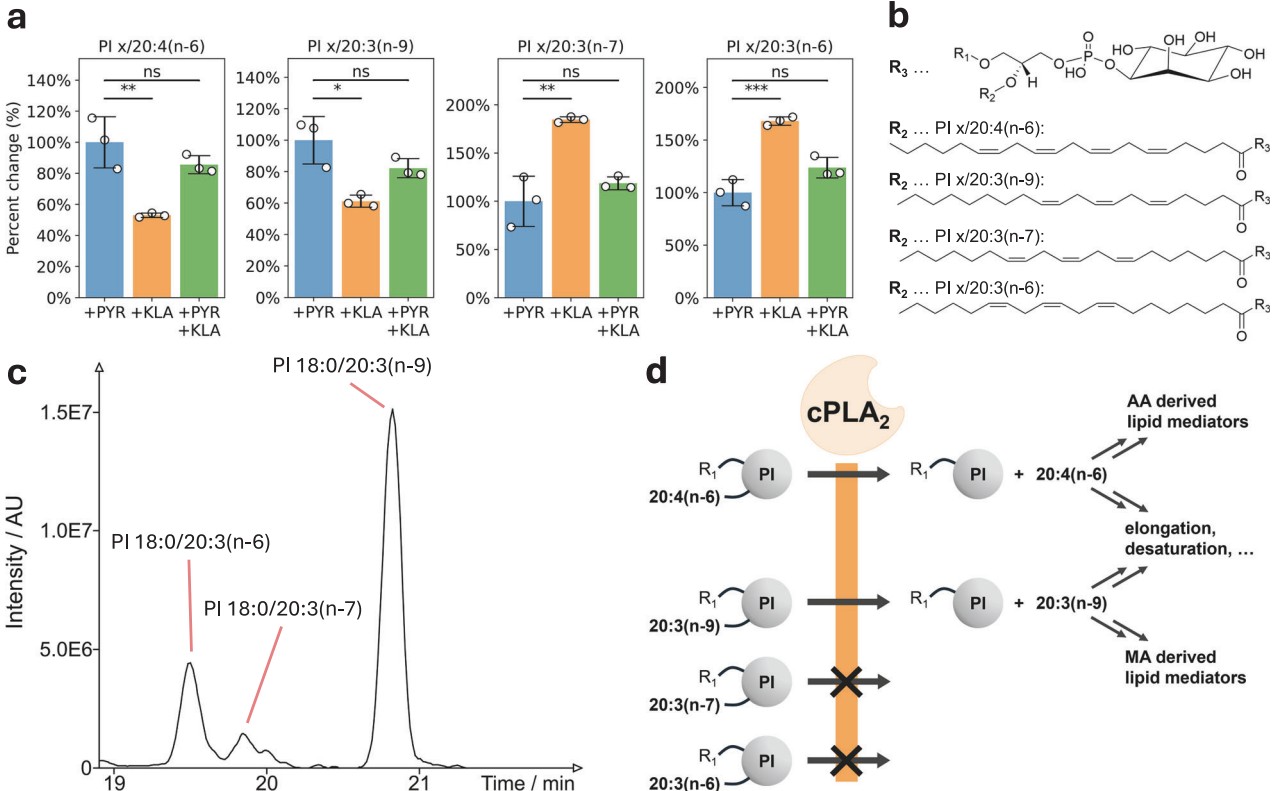

**Fig. 6 | cPLA₂ specificity toward FAs at the *sn*-2 position depends on the C=C positions. a** Assay conditions are as follows (n = 3 biological replicates each): (blue) control including the cPLA₂-specific inhibitor pyrrophenone (PYR) treatment, as prior work shows no significant difference from control lacking inhibitor[14]; (orange) cPLA₂ activation via stimulation with KDO₂ Lipid A (KLA)−the chemically defined version of the toll-like receptor-4 (TLR-4) activator lipopolysaccharide; (green) PYR pretreatment followed by 24 h KLA stimulation. Displayed values represent the combined changes in abundance (relative to +PYR) of all PI species with the respective FA at the *sn*−2 position, disregarding the FA at *sn*−1. Error bars represent the standard deviation. Shorthand notations represent the following significance thresholds for the two-sided statistical *t*-test: *** <0.001, ** <0.01, * <0.05, ns ≥ 0.05 (exact *p*-values from left to right,

first +PYR compared to +KLA, then +PYR compared to +PYR +KLA for each analyte: 0.0079, 0.2267; 0.0125, 0.1311; 0.0050, 0.2963; 0.0008, 0.0605). Source data is provided as a Source Data file. **b** Representative chemical structures of PI *x*/20:4(n−6), PI *x*/20:3(n−9), PI *x*/20:3(n−7), and PI *x*/20:3(n−6), featuring homo-allylic FAs, which are predominantly observed in higher organisms, and verified by EAD. **c** Experimental chromatogram of ω-positional isomers of PI 18:0/20:3, observed with a 30-min gradient. **d** Scheme of cPLA₂'s observed substrate specificity, including the previously discovered specificity towards AA and the C=C-position dependent specificity toward 20:3 isomers reported here. Increased amounts of downstream elongation and desaturation products further corroborate the finding about C=C position specificity of cPLA₂.

specificity by cPLA₂, are among the compounds reported in this study (Supplementary Data 2): PI 18:0/20:3(n−7) (Fig. 6, Supplementary Fig. 9b) and the observed downstream product of MA release by cPLA₂, PI 18:0/22:3(n−9) (Supplementary Figs. 7d and 9c).

The accuracy of LC=CL's ω-position assignments and its high sensitivity were confirmed in a benchmark with the orthogonal approaches EAD, PB, and OzID. Furthermore, the C=C positions of the identified ω-positional isomers of PI 18:0/20:3 (Fig. 6) were verified with EAD (Supplementary Fig. 9). Notably, EAD is only applicable to positive ion mode for the identification of lipid C=C positions[22], while several lipid classes, including PI, ionize much better in negative ion mode. To still achieve sufficient sensitivity using EAD for the verification of the three identified PI isomers, a tailored targeted MRMHR method was required (Methods). Moreover, we supplemented RAW264.7 cells with single FA precursors of the respective three C=C isomers (n−6, n−7, n−9) to further increase the likelihood of detecting C=C-specific fragments for those species. This highlights how LC=CL, due to its high sensitivity and applicability to both ion modes, can serve as a global screening approach to identify C=C isomers of interest. If information about additional C=C positions is required, they can be fully characterized through targeted fragmentation approaches such as MRMHR EAD. As such, LC=CL emerges as a powerful strategy for comprehensive, untargeted C=C-resolved lipidome

analysis, providing a detailed basis for selecting single candidates of interest for full structural characterization by tailor-made methods.

Embedded in the established open-source LDA software, LC=CL benefits from LDA's flexible decision rule concept, which facilitates platform independence by providing easy means to adapt it to any MS setup and fragmentation techniques, including emerging ion activation methods, such as EAD, UVPD, etc. The results of this study demonstrate the straightforward compatibility of LC=CL with complementary C=C localization techniques such as EAD. Such tandem approaches of LC=CL with emerging ion activation methods would offer full C=C characterization while benefiting from LC=CL's high sensitivity and chain-specific C=C information (Fig. 5b, c).

Notably, LDA supports the detection of a significantly larger number of compounds than those specifically investigated in this study, which was aimed at establishing the general method by focusing on phospholipids. As such, the LC=CL software is principally applicable to determining the C=C positions in the fatty acyl groups in triglycerides/diglycerides/monoglycerides, sphingolipids, sterol esters, acyl carnitines, etc. However, from a combinatorial perspective, the presence of more than two FAs exponentially increases the number of possible isomers. Thus, extensions to classes with more than two FAs warrant further investigations, particularly regarding chromatographic separation. If chromatographic separation is sufficient as for

the phospholipids shown in this study, LC=CL's RT-DB can be expanded to virtually any lipid classes and species, either by conducting further SIL-FA labeling experiments, e.g., with other organisms or SIL-FAs not included in this study, or also by complementary C=C position detection methods.

The RT-DB extension process is supported by a graphical user interface. Since the RT-DB is stored in an easily comprehensible Excel format, RT information from complementary methods can also be added straightforwardly by editing the file. While the RT-DB is already quite comprehensive, lipids featuring odd-chain FAs are lacking in the current RT-DB (only a single odd-chain FA (FA 21:4(n−6) has been identified, see Supplementary Fig. 10). This is due to the fact that all of our SIL-FA precursors included even chains, and mammalian organisms such as RAW264.7 cells regularly elongate FAs by two-carbon units[39]. While de-novo synthesis of odd-chain FAs is documented for mammalian cells[48] their abundance is significantly lower than that of even-chain FAs[49]. Due to the lower abundance and by using an untargeted DDA approach, MS/MS spectra were typically not acquired for these lipid compounds in the SIL experiments. Future studies involving strategies such as supplementation with odd-chain precursor FAs or emerging fragmentation techniques will enable the inclusion of these species in LC=CL's RT-DB.

According to the availability of graphical RT-DB extension tools dedicated to users without informatics background, and the lack of any apparent limitations in experimental MS/MS setups, we anticipate that the underlying RT-DB will grow steadily, fueled both by our contributions and those from the broader scientific community interested in large-scale, ω-resolved lipid identification. The latest version of the RT-DB is available at LCdbCL.github.io.

LC=CL allows for straightforward access to ω-position information, as demonstrated in the re-evaluation of the data already established in a previously published study[14]. This reanalysis revealed the specificity of $cPLA_2$ for MA, in addition to its well-established substrate, AA—a finding that would have remained concealed without ω-position information, due to the presence of other C=C positional isomers of 20:3. In RAW264.7 cells, MA is the second-most abundant FA in PI after AA[14]. The release of free MA by $cPLA_2$ is likely to have a profound impact on inflammation and lipid signaling, as MA is converted to various unique lipid mediators in the body[50]. The complementary insights provided by LC=CL will entail a deeper understanding of the biochemical mechanisms underlying enzyme regulations and their role in diseases. Accordingly, ω-position information made readily accessible by LC=CL will advance the field toward more authentic and accurate insights from a metabolomic perspective.

## Methods

### Cell culture
The murine RAW264.7 cell line was obtained from the American Type Culture Collection (ATCC #TIB-71). RAW264.7 cells were cultured as described elsewhere[14] in complete DMEM (containing 1% v/v P/S, 10% v/v FBS, and 1% v/v glutamine) in a humidified 37 °C incubator containing 5% $CO_2$. For FA supplementation experiments, log-phase RAW264.7 cells were plated at a density of ~1e6 cells/mL (7 mL for T-25 flasks, 20 mL for T-75 flasks) into complete DMEM containing natural or SIL FAs at 25 μM. The FAs were supplemented individually or in pairs of two at most to ensure adequate cellular uptake of each FA while keeping the complexity low, thus ensuring a high MS/MS spectra coverage. FAs used included $D_5$-18:3(n−3), $D_{11}$-18:2(n−6), $U$-$^{13}C$-16:1(n−7), $D_{17}$-18:1(n−9), $D_{19}$-16:1(n−10), 18:3(n−3), 18:2(n−6), 16:1(n−7), 18:1(n−9), and 16:1(n−10). The vehicle for FA supplementation was ethanol (final concentration of 0.06-0.08% v/v ethanol in complete DMEM). A control condition containing a vehicle only was also included. The cells were incubated for 24 h before collection. At collection, the flasks were approximately confluent.

### Lipid extraction, chromatography, and CID MS/MS analysis
The lipid extraction, chromatography, and CID MS/MS analysis used in this study were described elsewhere[14]. In short, RAW264.7 cells or human blood plasma samples were spiked with deuterated internal standards of phospholipids, either UltimateSPLASH ONE (USO) or EquiSPLASH (both from Avanti), extracted using the butanol:methanol [3:1] (BUME) method[51], and separated on a polar group end-capped solid-core reverse-phase column (CORTECS T3, 120 Å, 1.6 μm, 2.1 mm × 150 mm, Waters) using both 30-min and 60-min gradient methods, both starting at 75% buffer A (10 mM ammonium formate and 1% formic acid in water) to 100% buffer B (70/30 isopropanol/ acetonitrile with 10 mM ammonium formate and 1% formic acid) with a flow rate of 0.3 mL/min and a column temperature of 50 °C. The 30-min method starts at 25% solvent B from 0 to 0.5 min, ramps to 60% B at 2 min, ramps to 75% B at 7 min and holds until 15 min, ramps to 80% B at 22 min, ramps to 95% B at 26 min, ramps to 99% B at 33 min and holds until 34.5 min, ramps down to 25% B at 34.8 min and holds until 35 min. The 60-min method starts at 25% B from 0 to 1 min, ramps to 60% B at 4 min, ramps to 70% at 14 min, ramps to 75% B at 40 min, ramps to 99% B at 57 min holds until 59 min, then ramps down to 25% at 59.9 min and holds until 60 min. For CID analysis, a Vanquish UHPLC (Thermo Fisher Scientific) system was used in tandem with a Q-Exactive Hybrid Quadrupole-Orbitrap mass spectrometer (Thermo Scientific) with $MS^1$ resolution set at 70,000 (FWHM at $m/z$ 200) and $MS^2$ resolution set to 17,500. All samples of RAW264.7 cells and human plasma were measured in technical triplicate.

### MS/MS analysis with EAD
For EAD analysis, an Agilent 1290 Infinity II LC system (Agilent Technologies) with the same column and chromatographic conditions (30-min gradient) as for the CID analyses described above[14] was used in tandem with a ZenoTOF 7600 mass spectrometer (AB Sciex). The ion source settings on the ZenoTOF 7600 were as follows: ion source gas 1 at 50 psi, ion source gas 2 at 60 psi, collisional activated dissociation (CAD) gas at 7, curtain gas at 35, and a source temperature of 450 °C. A spray voltage of 5500 V was used. MS1 scans were performed over a mass range of $m/z$ 350–2000, with an accumulation time of 0.1 s, an electron kinetic energy of 12 eV, and a declustering potential of ±40 V.

IDA was performed using EAD mode at 12 eV and an electron beam current of 5500 nA. We chose 12 eV as this kinetic energy proved to be optimal for acquiring C=C position-specific EAD fragments in a ramping experiment (7–18 eV) using an in-house produced yeast lipid extract[52] (data not shown), which is also commercially available from Isotopic Solutions and Cambridge Isotope Laboratories. The accumulation time in MS/MS was set to 125 ms, with a reaction time of 30 ms and a Zenothreshold of 10,000 cps. An inclusion list with masses of known phospholipids was used with 5 maximum candidate ions and an intensity threshold of 100 ions/s, and dynamic background subtraction was applied.

The precursor masses of PC 38:3 & PC 38:4 ([M + H]+ & [M + Na]+), PE 38:3 & PE 38:4 ([M + H]+ & [M + Na]+) and PI 38:3 & PI 38:4 ([M + NH4]+) were selected for MRMHR experiments. Retention time scheduling was applied, and the accumulation times were adjusted for each compound (PC: 65 ms, PE: 95 ms, PI: 215 ms). C=C position-specific EAD fragments were automatically annotated by LDA using decision rule sets. The C=C positions were manually assigned by inspecting the corresponding spectra using LDA.

### Automated detection of SIL lipids with LDA
LDA performs automated quantification and annotation up to the lipid molecular species level. For this purpose, LDA utilizes a library of FAs (Excel file) for lipid molecular species level annotation (FA chain assignments). For the detection of SIL-FAs, this list was extended with all SIL-FAs potentially produced by RAW264.7 cells. To differentiate SIL-FAs from natural FAs, we added a prefix as encoding (single capital

letter) to the FA shorthand names[20], e.g., 'A18:1' and '18:1' correspond to the SIL-FA and natural FA, respectively. We used the following prefix encodings: 'A' corresponds to n−9 containing 17 D atoms, 'B' to n−6 ($D_{11}$), 'C' to n−3 ($D_5$), 'D' to n−7 ($^{13}C_{16}$) and 'E' to n−10 ($D_{19}$) (see Fig. 2b). Furthermore, LDA uses a comprehensive lipid species database (termed 'mass list'—also an Excel file) for automated quantitation/ annotation of lipid species. For streamlined generation of mass lists, we implemented a GUI (termed 'MassList Creator') in LDA that facilitates the automated generation of mass lists for user-definable lipid classes and adducts. When used for SIL lipid species, the MassList Creator generates all possible SIL lipid species (also, doubly labeled species and mixtures of different labels are supported). Moreover, we optimized LDA's 3D peak integration algorithm[31] toward increased peak separation capabilities to reliably distinguish between individual chromatographic peaks (different ω-positions). For distinguishing the isomeric lipid classes BMP and PG, authentic standards purchased from Avanti Polar Lipids were measured (Supplementary Fig. 11).

All obtained results conducted in biological and technical triplicates were manually verified to ensure high-quality RT-DBs, resulting in only 5% of detections identified as FPs. These were removed prior to generating the RT-DBs (Supplementary Data 4).

## Generation of RT-DBs

The first step of generating an RT-DB is the determination of the RTs of SIL phospholipid molecular species. For this purpose, LDA's standard routine for automated quantitation/annotation is used with the adaptations to SIL-FA species as described in the previous paragraph. Based on these data, the integrated LC=CL module extracts the corresponding RTs. While the impact of $^{13}C$ atoms on RT is negligible, deuterium isotope labels have a significant effect (Supplementary Fig. 12). Thus, the second RT-DB generation step corrects for this isotope effect on the measured RTs. For this purpose, unlabeled authentic standards (PC 16:0/20:4(n−6), PC 18:1(n−9)/16:0, and PC 18:1(n−10)/16:0), purchased from Avanti Polar Lipids, were measured in separate MS experiments. By comparing the RTs of the SIL species to the unlabeled authentic standards, LC=CL estimates $k_H$ and $k_D$ for the adjusted total isotope effect (aTIE)[53] for the respective measurement series (Eq. 1):

$$\text{aTIE} = \frac{(k_H/k_D) - 1}{f_a} + 1 \qquad (1)$$

where $k_H$, $k_D$, and $f_a$ correspond to the retention factor for protonated species, retention factor for deuterated species, and gradient-adjustment factor, respectively. $f_a$ is per default 1 but can be adapted to the respective gradient. Based on Eq. 1, the measured RTs of each deuterium-labeled lipid species are corrected with respect to the number of deuterium atoms in the compound. We observed that two SIL-FAs in the same lipid compound have an additive effect, i.e., the effect of each SIL-FA can be considered individually. Accordingly, LC=CL calculates the RT shift for each SIL-FA, and simply sums these shifts if there is more than one SIL-FA involved.

If combinations of unsaturated FAs pertaining to the same molecular species with different ω-positions result in the same RT after correcting for the isotope effect, LC=CL combines these identifications based on a user-definable grouping parameter (we used 5 seconds). To verify the validity of this supposition, we also conducted experiments by supplementing RAW264.7 cells with two SIL-FAs (different ω-positions). These experiments clearly indicated that an in silico aggregation of different ω-positions obtained in the same and in separate experiments is feasible: E.g., the individual identifications of PC 20:4 (n−6)_$D_5$−22:5(n−3), PC $D_{11}$−20:4(n−6)_22:5(n−3) and PC $D_{11}$-20:4 (n−6)_$D_5$-22:5(n−3), all with different SIL and thus different RTs prior to aTIE-correction were correctly automatically aggregated to PC 20:4(n−6)_22:5(n−3) as shown in Supplementary Fig. 1.

## LC=CL RT predictor

The LC=CL RT predictor maps the experimentally obtained RTs stored in the RT-DB to the actual RTs at the respective chromatographic conditions of the experiment. The RT predictor relies on anchor species derived from experimental data (detailed information about anchor species can be found in Supplementary Note 1). Importantly, the species utilized as anchors must be present for both, the original chromatographic conditions and the ones the RT-DB should be mapped to (see Supplementary Fig. 13). For this purpose, prior manual curation of biological data is not required, as the graphical user interface (GUI) offers a simple tool for visual anchor species selection (see Supplementary Fig. 14).

Typically, both the data of the original chromatographic conditions (reflected in the RT-DB) and the current experiments consist of several MS measurements. To derive the optimal anchor points, the LC=CL predictor provides three strategies depending on the availability of standards: (1) the number of anchor points is prioritized; (2) the reliability of anchor points is prioritized; and (3) all potential anchor points are matched. For data using standards, strategy #3 is applied; and for biological anchor species, #1 if standards are available to improve reliability, and #2 if these data lack a sufficient number of anchor species from standards.

Anchor points for each group (individual lipid classes, user-defined groups, or all available data combined) are binne,d and their elution profile mapping is performed via machine learning using a cubic spline interpolation model. The presented RT mapping algorithm is implemented based on the PolynomialSplineFunction class provided in the Apache Commons Mathematics Library, version 3.6.1. Further details regarding the LC=CL RT predictor can be found in Supplementary Note 1, including a benchmark using various chromatographic conditions (Supplementary Data 5) and sources of anchor points (Supplementary Data 6).

## Creation of a comprehensive RT-DB

We created individual RT-DBs from each measurement batch (RT-DB A30: biological replicate A, 30-min gradient; RT-DB B30a: first measurement of biological replicate B, 30-min gradient; RT-DB B30b: second measurement of biological replicate B, 30-min gradient; RT-DB C60: biological replicate C, 60-min gradient). Each of these measurement batches comprised five samples supplemented with the individual SIL FAs depicted in Fig. 2b and five samples supplemented with the respective unlabeled FAs to verify LC=CL's correction of the total isotope effect on RT (see Fig. 2c). Each sample was measured in triplicate. In addition, each measurement batch also includes individual measurements of unlabeled authentic standards (PC 16:0/20:4(n−6), PC 18:1(n−9)/16:0, and PC 18:1(n−10)/16:0) and the USO standard mix, purchased from Avanti Polar Lipids. Subsequently, we combined all experimentally verified ω-defined species obtained in the individual experiments into one comprehensive RT-DB. The LC=CL RT predictor was used to map between the chromatographic conditions (Supplementary Note 1 and Supplementary Fig. 15–17), and an in-house script combined all DB entries to a singular RT-DB, including information detailing in which measurement batch(es) the labeled molecular species were originally detected. We provide this aggregate RT-DB with the elution profile for the 30-min gradient (RT-DB_30min) and the 60-min gradient (RT-DB_60 min) used in this study, to ensure highly reliable RT mapping using the LC=CL RT predictor for different use-cases. To provide the scientific community with high-quality reference RTs specific for ω-positions in the aggregate RT-DBs, we manually curated occasional outlier RT values originating from the more challenging RT mappings between different gradients.

## Automated annotation of C=C positions

The result of the LC=CL RT mapping process is a new RT-DB Excel file that adheres to the established format of an LDA 'mass list',

complemented with ω-position-specific RT information. By using this RT-DB 'mass list', the MS measurements are processed by LDA's conventional quantification/annotation routine, and where possible, ω-positions are automatically assigned by the LC=CL annotation algorithm. The LC=CL annotation algorithm determines the accuracy of each potential C=C-position assignment based on the individual chromatographic peaks (see Supplementary Fig. 18). If no automated assignment is possible, the match will be provided as a suggestion, requiring manual attention in the LC=CL GUI (Supplementary Fig. 19). The number of fully automated ω-identifications by LC=CL correlates with the discriminative power of the experiment (Supplementary Fig. 4). For reliable separation of ω-position isomers, the degree of unsaturation is decisive: generally, for the discrimination of mono-unsaturated ω-position FA isomers, longer chromatographic gradients are needed than for polyunsaturated ω-position FA isomers. Moreover, the reliable annotation accuracy is further corroborated by experiments with unlabeled FA supplementation, where exactly those FAs increased quantitatively, whose ω-positions correspond to the supplemented ones (Supplementary Fig. 2).

### Benchmarking of LC=CL
For the analysis of NIST SRM 1950 human plasma with LC=CL as well as with EAD, the experimental protocol for lipid extraction, chromatography, and MS/MS analysis described above was employed. Twenty of the LC=CL identifications were derived from ω-position evidence provided by the EAD measurements (see Supplementary Data 3). These were added to the employed RT-DB for analysis with LC=CL. Of note, while NIST SRM 1950 human plasma was analyzed with LC=CL, EAD, and OzID, a different human plasma sample was profiled in the PB studies. While EAD results coincided with all of LC=CL's ω-position assignments, an excellent agreement of 96,7% of the ω-position identifications was also achieved with PB and LC=CL for lipid molecular species identified in common.

### Analysis of cPLA$_2$ assays with LC=CL
Analysis followed the procedures outlined in the previous paragraphs. Thereafter, lipid identifications were quantified relative to the EquiSPLASH™ internal standard mixture from Avanti Polar Lipids. From LDA's options for standardization[31], we chose 'relation to protein content'. The significance of the relative change in abundance was determined using a two-sided $t$-test, while assuming that the assay conditions exhibit similar variances. No data were excluded from this analysis.

### Data processing
The LDA results (version 2.11.0, includes the LC=CL module—see Code availability) were further processed with Python 3.6. Postprocessing of the LDA output, including further statistical analysis, was carried out using pandas 2.2.2[54], numpy 1.26.4[55], statsmodels 0.14.2[56], and scipy 1.13.1[57]. Data was visualized using matplotlib 3.10.0[58] and seaborn 0.13.2[59].

### Reporting summary
Further information on research design is available in the Nature Portfolio Reporting Summary linked to this article.

## Data availability
Raw and processed experimental data used in the creation of all RT-DBs in this study, as well as the validation and benchmarking of the LC=CL algorithm, including EAD data, have been deposited in the GNPS public repository MassIVE[60] under the accession code MSV000097637 [https://doi.org/10.25345/C5CV4C43V][61]. Additionally, we provide SIL-FA chain libraries, SIL-FA mass lists, optimized 3D parameters, and decision rule sets extended for SIL complexity and the comprehensive aggregate RT-DBs at the same MassIVE location. As recommended by

the International Lipidomics Society, lipidomics minimal reporting checklists[62] are provided (https://doi.org/10.5281/zenodo.15719444 and https://doi.org/10.5281/zenodo.15719466). Source data are provided with this paper.

## Code availability
LDA version 2.11.0 featuring the LC=CL module, including the SIL-FA chain libraries, SIL-FA mass lists, optimized 3D parameters, and decision rule sets extended for SIL complexity, is freely available from [http://genome.tugraz.at/lda2/]. The source code is released under a GNU GPL v3 license and is available from [https://github.com/ThallingerLab/LDA2/][63].

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

## Acknowledgements

We thank Ulrich Stelzl for many fruitful and insightful discussions about the project. The authors gratefully acknowledge the financial support of the University of Graz. J.H. was partially funded by a Max Kade fellowship awarded by the Austrian Academy of Sciences. Work on this project at the University of California, San Diego was supported by the U.S. National Institutes of Health NIGMS MIRA grant R35 GM139641, which is a renewal of RO1 GM20501-44 (E.A.D.).

## Author contributions

L.M.L., J.P.A., O.Q., E.A.D., and J.H. conceptualized the project. J.H., E.A.D., O.Q., and E.R. contributed to funding and resources. L.M.L. designed and implemented the LC=CL algorithm and GUI, curated the RT-DBs, and performed the data analysis. L.M.L. and J.H. adapted LDA for stable isotope-labeled lipids. J.P.A. and G.M.M. carried out the cell culture experiments. The RPLC-MS/MS experiments were performed by J.P.A., G.M.M., and A.M.A. using the Q-Exactive, and using the ZenoTOF by M.P. and E.R., including the implementation of IDA and MRMHR methods for the EAD fragmentation on samples provided by G.M.M. R.A.T. and L.M.L. searched in the public databases for already reported species. The data analysis of the EAD measurements was performed by L.M.L. and J.H. BMP and PG standards were measured by G.M.M. L.M.L. and J.H. wrote the initial draft. G.M.M. and E.A.D. did extensive editing, and all co-authors participated in the review and revision of the paper.

## Competing interests

The authors declare no competing interests.
