## [Transparent Peer Review file · Nature Communications]

Computationally unmasking each fatty acyl C=C position in complex lipids by routine LC-MS/MS lipidomics

Corresponding Author: Professor Jürgen Hartler

Version 0:

Reviewer comments:

Reviewer #1

(Remarks to the Author)

The authors present LC=CL, a software feature within the LDA software to identify the omega-position in desaturated fatty acyls. This feature is mainly based on a retention time-data base (RT-DB) since different omega-species are differentiated on the chromatographic level by different retention times. The presented feature has several advantages. To me the most impact is generated by the possibility to identify omega-species without the need for bigger investments in order to apply specialized experimental techniques and MS-setups such as EAD which has become very popular in recent year with instruments such as the ZenoTOF. However, there are a few issues that I recommend to address to further increase the quality of the manuscript.

1. It would be interesting to compare LC=CL based analysis results to EAD-based ones to illuminate pros and cons of each method.
2. The Discussion section is currently not a discussion as it doesn't contain any critical issues, comments on limitations or potential biases, as well as an embedding in / comparison with other techniques.
3. In my view, the selection of figures is suboptimal. The concept figures, while giving an overview, are not very informative. The most information is contained in Figure 6. I'd recommend to revisit the overall figures concept and their core messages. In general, I think readers would appreciate one or several figures that specify the conceptual steps shown in Fig. 2 summarizing the subsections of the methods section.
4. To me the paper is clearly a method paper describing the LC=CL feature. The authors also mention a discovery type result when they report that some lipid species have been found in murine macrophages for the first time (Fig. 3). I recommend to make a results subsection dedicated to this discovery increasing the overall focus and clarity of the paper.

The paper is overall well written but could be better structured to increase clarity and conciseness. Regarding the actual LC=CL feature and the RT-DB as a resource I think this paper is a valuable addition for the lipid/lipidomics community.

(Remarks on code availability)

Reviewer #2

(Remarks to the Author)

This manuscript reports an attempt of using retention time information from routing RPLC-MS/MS to identify the C=C locations in lipids. Stable isotope labeled fatty acids were fed to cells to produce isotope labeled lipids, which were subsequently analyzed to generate the data base. The authors claim that a total of 2408 unsaturated phospholipid species were identified, with 1138 reported for the first time.

Fast identification of C=C location isomers is of a great significance. However, the existence of differences, for a large number of C=C isomers, that are significant enough for algorithm training and subsequent identification of unknown isomers, is really against the common knowledge for RPLC-MS/MS analysis of lipids. For example, have the authors used another method to confirm the structure of the lipids and the C=C locations of the lipids shown in Figure 6b; how many C=C location

isomers with confirmed structures could really show this kind of differences in retention time? Usually, for most phospholipids, the C=C location isomers could barely show any difference in retention time when being analyzed by RPLC-MS/MS. Predicting a coverage more than what have been previously identified experimentally does not serve as a sufficient justification.

A systematic validation is required indeed before this concept could be used as a valid assumption and base for the work reported in this manuscript.

Another comments, even if the fundamental based of this method was validated, the authors should also address the isotope effects in physiology (as indicated by some previous work done by the co-authors from UC San Diego), which might impact the practical application of this method.

(Remarks on code availability)

Reviewer #3

(Remarks to the Author)

The work of Lamp et al is a great example of both a fundamental and applied investigation towards utilization of routine LC-MS/MS lipidomics for assignment of double bond positions in various lipid classes. The localization of C=C positions in unsaturated complex lipids is a current and major focus of research, and growing evidence demonstrates their critical role in various physiological and pathological conditions. Therefore, the topic is of broad interest and not only limited to researchers in the field of lipidomics. The reviewer was very impressed with the authors dedication not only to the fundamental analysis of these compounds from both an instrumental and computational perspective, but also in the applied workflow of this pipeline for analysis of real-world complex samples to demonstrate its utility. The manuscript is well written with straightforward visuals, and the Supporting Information is very detailed and informative. In this reviewer's opinion, the work is suitable for publication in the prestigious journal Nature Communications. However, to improve the already high quality even further, some revisions are suggested, which are listed below:

1. Abstract: Please name the used cell line (RAW264.7) and give a short rationale why this was selected. The selection should be addressed also in the main manuscript in some more detail. The authors report 1,138 newly reported compounds. Are these new in general, or new for the investigated cell line? For me, it is not fully clear if these are due to the feeding experiments with different fatty acids. Please clarify in the manuscript.

2. Figure 6C: The peak of PI 18:0/20:3(n-7) at roughly RT 20 min is not fully resolved. Do you have an idea what this peak might be? A different DB location for example, or an interfering m/z from a different lipid? How does the software deal with such unresolved chromatographic peaks?

3. The authors state (line 265ff): "LC=CL is equally applicable to determining the C=C positions in the fatty acyl groups in triglycerides/diglycerides/monoglycerides, sphingolipids, sterol esters, acyl carnitines, etc." Did the authors also investigate triglycerides? Due to the bound 3 fatty acyls, I expect due to more possible SN-positions also more isomers and thus, more chromatographic peaks which might not be resolved. Do you have already data on TAGs?

4. A further important glycerophospholipid class are bis(monoacylglycero)phosphates (BMP). This lipid class should also be present in RAW264.7 cells, albeit in minor concentrations. Were BMPs detected? Are these PG-regioisomeric lipids also covered by the methodology? Please address this aspect also in the manuscript.

5. Please add a discussion on the limitations of the method (see points above). It would be a valuable information if the method is (up to now?) restricted to lipids with 2 bound fatty acids, or if it is working as nicely also for lipids incorporating 3 fatty acyls (e.g. TAGs) or even 4 fatty acyls (like the important glycerophospholipid class cardiolipin)?

Editorials:

- line 305: Please specify the used HPLC columns
- line 304: typo in "Quadrapole"
- Supp. Info: Figure 2: Please specify "Lyso"

(Remarks on code availability)

Version 1:

Reviewer comments:

Reviewer #1

(Remarks to the Author)

The authors significantly modified the manuscript which has improved a lot in the process. All reviewers' comments, not only

mine, have been adequately addressed. However, a few small issues now remain regarding the newly provided information:

1. Thank you for adding a comprehensive comparison of LC=CL with experimental methods such as OzID and EAD. The latter has not been around for a very long time and there have been a few differences across analysis software applications for EAD fragment spectra. As indicated the EAD analyses were performed in the Rampler lab in Vienna. Please indicate if a software such as MS-DIAL 5 was used for automatic identification of the EAD spectra or whether they were manually analyzed.

2. The NIST plasma EAD mass spectral data would be a great resource for the lipid (computational) community and I would strongly encourage to publish it with this paper.

3. eV optimization is highly important in EAD as slight changes (10 eV or 15 eV) cause vastly different fragmentation. Please indicate whether 12eV was selected after ramping or whether it was selected based on previous experience.

I know this manuscript isn't supposed to focus on EAD but to get a sufficient impression of the EAD data and results the requested information is necessary.

Overall, once these minor issues have been addressed, I recommend the manuscript for publication.

(Remarks on code availability)

Reviewer #2

(Remarks to the Author)

The authors have added more results and details in the revised manuscript. However, the main concern remains unresolved. The key issue lies in how the identity of each lipid species in the mixture produced by the cells is characterized using mass spectrometry. It appears that the identification of lipid C=C location isomers relies on successful LC separation of each isomer. However, this approach may not be effective, as separating most phospholipid isomers is extremely challenging, even with optimized RPLC methods.

The concern regarding Fig. 6 also remains unaddressed:

1. The chromatogram shows three peaks; however, with RPLC, many phospholipid species with similar fatty acyl/alkyl chain lengths are expected to co-elute in this retention time window. Thus, assigning these three peaks to the n-6, n-7, and n-9 isomers of PI 18:0/20:3 is unconvincing.

2. The authors present EAD-MS analysis of these peaks in Supplementary Fig. 9. However, most of the peaks are barely distinguishable from background noise, making the identification of C=C locations unreliable. Furthermore, co-eluting phospholipids at this retention time are likely to generate overlapping signals, which could lead to mischaracterization of the lipid species.

(Remarks on code availability)

Reviewer #3

(Remarks to the Author)

The authors did an excellent job in addressing all my comments adequately. The described methodology is original and represents an important contribution to lipidomics studies down to the double bond positional level. Therefore, to my opinion, publication in Nature Communications is justified after clarification the following minor issue.:

In the main manuscript, page 9, line 219ff it is stated: "In the provided example, PC 18:0/22:5(n-3) coelutes with PC 18:0/24:5(n-6). Such coeluting isomers are readily identified by the LC=CL with high confidence, as the algorithm assigns ω -positions only when MS/MS evidence is available for lipid molecular species."

Should it not be the isomer PC 16:0/24:5(n-6) instead?

And in line 222ff "As this example illustrates, knowledge of the sum composition alone, i.e. PC 38:5, does not suffice for an assignment of C=C positions based on RT."

Should this not be "PC 40:5" for consistency?

(Remarks on code availability)

We thank all of the reviewers for their serious engagement in reading our manuscript and providing constructive and encouraging feedback. We have addressed all of the reviewer's concerns as well as adapted/extended the manuscript, which has improved the paper considerably. Changes in the manuscript and the supplement are indicated in green color.

Reviewer #1

- 1) *It would be interesting to compare LC=CL based analysis results to EAD-based ones to illuminate pros and cons of each method.*

We have now included a benchmark with EAD. Data of NIST SRM 1950 human plasma has been acquired in IDA mode at the Rampler lab in Vienna (Fig. 5). The EAD results confirmed 100% of our LC=CL's ω -position assignments for the lipid molecular species identified in common between LC=CL and EAD, where C=C specific fragments were present in EAD spectra (see Supplementary Table 3). We elaborated on pros and cons of each method in the subsection 'Benchmarking of LC=CL with EAD, PB and OzID on human plasma' (page 7) and in the Discussion (p. 10).

- 2) *The Discussion section is currently not a discussion as it doesn't contain any critical issues, comments on limitations or potential biases, as well as an embedding in / comparison with other techniques.*

Thank you for raising this point. We have now significantly expanded the Discussion (p. 10 and 11) to include comparisons with other techniques, limitations such as the need for further investigations particularly regarding chromatography to reliably apply the approach to lipid classes with more FAs such as TGs.

- 3) *In my view, the selection of figures is suboptimal. The concept figures, while giving an overview, are not very informative. The most information is contained in Figure 6. I'd recommend to revisit the overall figures concept and their core messages. In general, I think readers would appreciate one or several figures that specify the conceptual steps shown in Fig. 2 summarizing the subsections of the methods section.*

We agree that the core concept could have been presented in a more comprehensive and more informative way. Accordingly, we revisited Fig. 2 to now also include the analytical workflow for generating an RT-DB as described in subsection 'Creation of ω -position resolved retention time database (RT-DB)'. Fig. 3 was added to more easily convey the content of the subsection 'Machine learning-based RT-DB mapping'. The previous Figures 3 and 4 were combined to Fig. 4 to summarize the experimental outcomes more concisely. The results of the benchmark with EAD are now visualized in Fig. 5.

- 4) *To me the paper is clearly a method paper describing the LC=CL feature. The authors also mention a discovery type result when they report that some lipid species have been found in murine macrophages for the first time (Fig. 3). I recommend to make a results subsection dedicated to this discovery increasing the overall focus and clarity of the paper.*

Thank you for this suggestion to increase the clarity of the paper. To separate the methodological part from the experimental results more clearly, we have now introduced a subsection titled 'Results of stable isotope labeling experiments' (p. 6 and 7).

- 5) *The paper is overall well written but could be better structured to increase clarity and conciseness.*

Excellent point. We restructured the manuscript according to your suggestions.

Reviewer #2

1) *Fast identification of C=C location isomers is of a great significance. However, the existence of differences, for a large number of C=C isomers, that are significant enough for algorithm training and subsequent identification of unknown isomers, is really against the common knowledge for RPLC-MS/MS analysis of lipids. For example, have the authors used another method to confirm the structure of the lipids and the C=C locations of the lipids shown in Figure 6b; how many C=C location isomers with confirmed structures could really show this kind of differences in retention time? Usually, for most phospholipids, the C=C location isomers could barely show any difference in retention time when being analyzed by RPLC-MS/MS. Predicting a coverage more than what have been previously identified experimentally does not serve as a sufficient justification. A systematic validation is required indeed before this concept could be used as a valid assumption and base for the work reported in this manuscript.*

Thank you for highlighting the significance of our approach for fast identification of C=C location isomers. First of all, we want to clarify that we report only species that were experimentally verified by stable isotope labeling experiments. As such, the underlying retention time database forms an experimentally verified resource of the elution profile of C=C isomers of phospholipids. We do not train any algorithm to predict the ω -position of any identifications that could not be unambiguously matched to the experimentally verified ones in the database. We are sorry, if that was not clear in the previous version of our manuscript. We now made a pertinent statement to this effect in the subsection 'Machine learning-based RT-DB mapping' (page 5, before Fig. 3).

Second, to match the retention times in the database to the changing chromatographic conditions, we use the cubic spline interpolation algorithm, which is particularly suited for this purpose. We now provide a rationale for the choice of this algorithm and its high accuracy in the subsection 'Machine learning-based RT-DB mapping' (p. 5).

Third, we demonstrate that the employed chromatography can separate C=C location isomers with several validations. We appreciate the referee's insight to the fact that previously published studies in this field have not achieved the resolution that is reported in our manuscript. We trust that with our responses to his/her concerns, the referee can recognize the advances in this subject area that this manuscript reports. We now try to make this clearer at several points in the manuscript (see below). Overall, the validity of the whole approach has been experimentally verified by the following six orthogonal verifications:

1. The labeling experiments verify that the C=C resolved species actually exist, and also their elution profile – this verifies the validity of the database (see 'Creation of ω -position resolved retention time database (RT-DB)', pages 3-5).
2. Experiments with RAW264.7 cells verified that the RT-DB can be mapped to other chromatographic conditions, including different mobile and stationary phases, and also different gradients (see Supplementary Note 1).
3. Analysis of data from RAW264.7 cells supplemented with pairs of native (without stable isotope labels) FAs allowed for the fully automated assignment of 2074 lipid identifications with resolved ω -positions. Quantitative evaluation of these data revealed that the overall intensity of species with respectively assigned ω -positions increased in comparison to not supplemented RAW264.7 cells, while the intensity of species with other assigned ω -positions showed a relative decrease (Supplementary Fig. 2). This experiment verifies that the correct assignments were made, and as such the validity of the whole workflow, including that the chromatography can clearly separate the different C=C location isomers. We apologize that we

did not describe this purpose of the experiment more clearly in the previous version of the main text. We have now added a more pertinent statement referencing Supplementary Fig. 2 in the subsection 'Creation of ω -position resolved retention time database (RT-DB)' on p. 5.

4. The validity of LC=CL's ω -position assignments were confirmed in a benchmark to published results of two orthogonal C=C localization methods, i.e., OzID and Paternò-Büchi (see Supplementary Table 3), both analyzing human plasma. These results indicate excellent agreement.
5. Rerunning the same NIST plasma used for the original benchmarking of LC=CL (see point #4) on a ZenoTOF7600 at the Rampler lab in Vienna, using EAD acquired in IDA mode, confirmed 100% of the ω -position assignments for lipid molecular species identified in common, where C=C specific fragments were present in EAD spectra (see Supplementary Table 3). From this evaluation, it is clear that the sensitivity of EAD is much lower than the one of our LC=CL approach (see Fig. 5). This experiment not only verified the validity of the whole approach, but both, that the used chromatography for separating C=C isomers can be reproduced anywhere (see also Supplementary Fig. 3b), and that the method can be used in combination with any fragmentation technique.
6. To verify the structure of the lipid species shown in Fig. 6b, we now performed MRMHR EAD experiments on a ZenoTOF7600. For this purpose, we supplemented RAW264.7 cells with single FAs of the respective three C=C isomers (n-6, n-7, n-9) to increase the probability of acquiring C=C specific fragments for those species, as PI ionizes much worse in positive ion mode (but positive ion mode is required for C=C position detection with EAD (DOI: [10.1021/acs.analchem.5b01460](https://doi.org/10.1021/acs.analchem.5b01460))). To reveal the C=C-specific fragments, a tailor-made targeted MRMHR method was required (see Methods). Similar to the analysis alluded to in point #3 for obtaining the global phospholipidome of RAW264.7 cells, the MS¹ intensity profile showed a clear increase of the species with the respective ω -position for the three species (see Supplementary Fig. 3a). Moreover, all C=C positions of all three C=C isomers could be identified, confirming LC=CL's assignments (see Supplementary Fig. 9). Besides the confirmation of our claims, this verification also demonstrated the severe limitations of other C=C location identification methods in untargeted profiling approaches. Often, specialized targeted setups are required to reveal the same as LC=CL would deliver in an untargeted fashion.

In summary, the validity of our approach has been demonstrated by six different methods, four of which successfully verified the overall concept with stellar agreement. We hardly know of any published results whose validity has been scrutinized to such an extent. Due to the limited sensitivity of other C=C localization methods and, as demonstrated in point #6, the severe difficulties to verify single low abundant species by them, further verifications are beyond the generally acceptable high requirements for introducing a new method. Moreover, the EAD experiments (points #5 and #6) brilliantly demonstrate that LC=CL can be easily combined with any other orthogonal fragmentation-based C=C localization methods. Specifically, if the evaluation of a biological study with LC=CL revealed that a particular C=C isomer is of special interest (such as the one shown in Fig. 6), specialized targeted setups (such as MRMHR) in combination with LC=CL could be used to fully resolve all C=C positions. As such, the LC=CL can be seen as the ultimate approach for untargeted C=C resolved global screening of the lipidome, which forms the base for more detailed targeted analyses.

- 2) *Another comments, even if the fundamental based of this method was validated, the authors should also address the isotope effects in physiology (as indicated by some previous work done by the co-authors from UC San Diego), which might impact the practical application of this method.*

Although SIL were used to identify lipid ω -positions for establishing the RT-DB, once established the software is applied to native samples without labels. Only FAs with deuterium labels at the enzymatically inert sites after the terminal C=C were chosen for the current study. As outlined in the subsection 'Creation of ω -position resolved retention time database (RT-DB)' (p. 3 and 4), this ensured that the SIL is not affected by β -oxidation, elongation and desaturation, as these processes occur at the carboxy terminus of FAs. Moreover, also ^{13}C labels, as present in $^{13}\text{C}_{16}$ -16:1(n-7) are not affected by elongation and desaturation. The SIL in the FAs used for the study by Navratil AR, Shchepinov MS, and Dennis EA (2018) "Lipidomics reveals dramatic physiological kinetic isotope effects during the enzymatic oxygenation of polyunsaturated fatty acids ex vivo" *J Am Chem Soc*, **140**, 235-43. (DOI: [10.1021/jacs.7b09493](https://doi.org/10.1021/jacs.7b09493)) were on purpose located at the bis-allylic sites in the FAs, as these are the hydrogen/deuterium atoms abstracted by the enzymes in the synthesis of eicosanoids. In the present case, all deuteration sites were chosen to be inert to normal enzymatic reactions, which is the opposite purpose from that employed in the Navratil paper.

Reviewer #3

- 1) *Abstract: Please name the used cell line (RAW264.7) and give a short rationale why this was selected. The selection should be addressed also in the main manuscript in some more detail.*

This is an excellent suggestion. Now, we explicitly name the cell line in the abstract. However, due to word limitations for the abstract, we preferred to provide the rationale in the main text only – in the subsection “Creation of ω -position resolved retention time database (RT-DB)” on page 3. We chose this cell line because RAW264.7 cells express a wide variety of desaturation enzymes as outlined by Xia, T. *et al.* (DOI: [10.1016/j.jlr.2023.100410](https://doi.org/10.1016/j.jlr.2023.100410)).

The authors report 1,138 newly reported compounds. Are these new in general, or new for the investigated cell line? For me, it is not fully clear if these are due to the feeding experiments with different fatty acids. Please clarify in the manuscript.

These 1,138 (now 1,145 as explained in the answer to comment #4) ω -position-resolved novel compounds have not been reported for any other organism within major lipid databases before. We clarified the statement in the caption of Fig. 4 and the corresponding text below (page 7) accordingly.

We are sorry if it was not clear in the previous version that one major factor for this high number is that the feeding experiments increase the production of low abundant lipid species, and as such the probability to detect them. The second major factor is that the sensitivity of the chosen LC-MS/MS setup remains uncompromised by our approach, as we do not require any derivatization or fragmentation techniques with a low yield of C=C position-specific ions. To emphasize these two important points, we added a pertinent statement to the Discussion (p. 10, third paragraph of the Discussion) providing a rationale for the high number of novel compounds.

- 2) *Figure 6C: The peak of PI 18:0/20:3(n-7) at roughly RT 20 min is not fully resolved. Do you have an idea what this peak might be? A different DB location for example, or an interfering m/z from a different lipid? How does the software deal with such unresolved chromatographic peaks?*

Unfortunately, no MS/MS spectra were acquired for the minor peak at roughly RT 20 min. Accordingly, there is no MS/MS evidence for what the corresponding species might be. To showcase how LDA deals with unresolved chromatographic peaks of coeluting FA isomers, we now included an example in Fig. 5. In such a case, the obtained MS¹ quantity is split according to the distinct fragments, as described in DOI: [10.1038/nmeth.4470](https://doi.org/10.1038/nmeth.4470). LDA features an algorithm for detecting overlapping MS¹ signals and isotopic peaks to reduce their effects (DOI: [10.1093/bioinformatics/btq699](https://doi.org/10.1093/bioinformatics/btq699)). If the chromatographic method is not able to resolve C=C-positional isomers of the same lipid molecular species, then LC=CL does not assign ω -positions, but instead provides suggestions that the user may select from (see Supplementary Fig. 4 as well as the description in the subsection ‘Automated annotation of C=C positions’ in the Methods on p. 15).

- 3) *The authors state (line 265ff): “LC=CL is equally applicable to determining the C=C positions in the fatty acyl groups in triglycerides/diglycerides/monoglycerides, sphingolipids, sterol esters, acyl carnitines, etc.” Did the authors also investigate triglycerides? Due to the bound 3 fatty acyls, I expect due to more possible SN-positions also more isomers and thus, more chromatographic peaks which might not be resolved. Do you have already data on TAGs?*

While the LC=CL software also supports lipid classes with more than two FA chains, RAW264.7 cells are not ideal for the investigation of TAGs as this cell line does not produce a high variety of TAGs and

barely incorporates any PUFAs in this lipid class. To cover TAGs, we would have to repeat the experiments with another more appropriate cell line, which is beyond the scope of this manuscript. We now provide a detailed explanation of potential challenges that might arise in an extension of our method to lipid classes with more than two FAs (p. 11).

- 4) *A further important glycerophospholipid class are bis(monoacylglycero)phosphates (BMP). This lipid class should also be present in RAW264.7 cells, albeit in minor concentrations. Were BMPs detected? Are these PG-regioisomeric lipids also covered by the methodology? Please address this aspect also in the manuscript.*

Thank you for raising this important point that we would have overlooked. We purchased BMP and PG standards from Avanti Polar Lipids. Based on these data (Supplementary Fig. 10), we were able to correctly assign the BMP species in our RT-DB that were previously misidentified as PG. Accordingly, we also repeated the searches of major lipidomics databases with this corrected information, which resulted in a small difference in reported novel species (previously 1,138, now 1,145). We updated Figures 4b and c correspondingly. Of note, the number of novel species where ω -positions were omitted decreased from 132 to 87 species, as the databases have been updated since our last search.

- 5) *Please add a discussion on the limitations of the method (see points above). It would be a valuable information if the method is (up to now?) restricted to lipids with 2 bound fatty acids, or if it is working as nicely also for lipids incorporating 3 fatty acyls (e.g. TAGs) or even 4 fatty acyls (like the important glycerophospholipid class cardiolipin)?*

We added a discussion on the limitations of the method in the Discussion (p. 11), emphasizing that lipid classes with more than 2 FAs are supported from a software's perspective, but further investigations/optimizations are required to clarify whether the chromatographic separation is sufficient (see also answer to #3).

- 6) *Editorials*

Thank you very much for making us aware of them. We addressed/corrected them correspondingly in the manuscript and the supplement.

We thank all of the reviewers for their thorough engagement and efforts in assessing the value of our revised manuscript. We have addressed all of the reviewers' concerns and adapted/extended the manuscript and the supplement where appropriate.

Reviewer #1

- 1) *Thank you for adding a comprehensive comparison of LC=CL with experimental methods such as OzID and EAD. The latter has not been around for a very long time and there have been a few differences across analysis software applications for EAD fragment spectra. As indicated the EAD analyses were performed in the Rampler lab in Vienna. Please indicate if a software such as MS-DIAL 5 was used for automatic identification of the EAD spectra or whether they were manually analyzed.*

While we indicated in the LC=CL guide for reviewers that LDA was used for this purpose, we did not mention it in the manuscript. Thank you for bringing this to our attention. We developed decision rules for the automated annotation of the C=C position-specific EAD fragments, which are available on the MassIVE repository (<https://doi.org/10.25345/C5CV4C43V> – see LCCL_ReviewerGuide for login details) and also in the LDA installation directory in the subfolder 'fragRules/ ZenoTOF7600/EAD_12eV_pos/'. These rules will be released with LDA version 2.11 at the time of publication of LC=CL. While LDA can automatically annotate C=C position-specific EAD fragments, it does not derive C=C position information from MS/MS fragments, as this feature is not yet implemented in LDA. The C=C positions were manually assigned by inspecting the corresponding spectra using LDA. We added a pertinent statement to the "MS/MS analysis with EAD" in the Methods section (p. 9).

- 2) *2. The NIST plasma EAD mass spectral data would be a great resource for the lipid (computational) community and I would strongly encourage to publish it with this paper.*

Thank you for this suggestion. We have already uploaded all NIST EAD data to the MassIVE repository (<https://doi.org/10.25345/C5CV4C43V> – see LCCL_ReviewerGuide for login details) in the previous revision. You can find the raw data in the subfolder 'raw/RAW_Files/NIST_SRM_1950_EAD/' and the LDA annotations in the subfolder 'search/LDA_Results/NIST_SRM_1950_EAD/'. For visualizing the annotated spectra, the respective chrom folder and LDA's result Excel file have to be downloaded. Details how to use the LDA to visualize spectra can be found in chapter 5 and 5.4 of LDA's user manual (https://genome.tugraz.at/lda2/2.10/LDA_2.10.pdf).

- 3) *eV optimization is highly important in EAD as slight changes (10 eV or 15 eV) cause vastly different fragmentation. Please indicate whether 12eV was selected after ramping or whether it was selected based on previous experience.*

Thank you for pointing this out. The kinetic energy was optimized using an in-house produced yeast lipid extract, ramping from 7 eV to 18 eV. This extract serves as our internal benchmarking material for lipidomics experiments, also containing phospholipids (<https://doi.org/10.1039/C7AN00107J>). The extract is also commercially available from Isotopic Solutions and Cambridge Isotope Laboratories in both labeled and unlabeled formats. In this study, we used the yeast extract to assess routine instrument performance and optimize EAD fragmentation parameters. This optimization indicated that 12 eV is ideal for acquiring C=C position-specific EAD fragments. This observation aligns with reports from other laboratories and Sciex application notes, where 12 eV is commonly used for EAD-based lipidomics on the ZenoTOF platform (see https://sciex.com/content/dam/SCIEX/pdf/posters/amer/asms2021/lifescienceresearch/950AM_Oral_Thursday_Pearson.pdf, slide 15). We have clarified this in the manuscript by adding a corresponding note to the "MS/MS analysis with EAD" section in the Methods section (p. 9).

- 4) I know this manuscript isn't supposed to focus on EAD but to get a sufficient impression of the EAD data and results the requested information is necessary. Overall, once these minor issues have been addressed, I recommend the manuscript for publication

We appreciate that this reviewer recognizes that this manuscript is on LC=CL and that the EAD data was only included because of reviewers' requests for EAD verification of the C=C position assignments. The EAD-related information requested by the reviewer is provided in the Methods section, and EAD data is available at the MassIVE repository.

Reviewer #2

- 1) *The authors have added more results and details in the revised manuscript. However, the main concern remains unresolved. The key issue lies in how the identity of each lipid species in the mixture produced by the cells is characterized using mass spectrometry. It appears that the identification of lipid C=C location isomers relies on successful LC separation of each isomer. However, this approach may not be effective, as separating most phospholipid isomers is extremely challenging, even with optimized RPLC methods.*

Thank you for reviewing our manuscript very critically. We would like to clarify that our method does not require chromatographic separation of all FA chain isomers – only the ω -position isomers need to be separated, which is indeed the case. This is explicitly stated in the manuscript (see “Machine learning-based RT-DB mapping” and “Supplementary Note 1”):

“In comparison to that, ω -position isomers were typically separated by ten seconds or more (Supplementary Fig. 4).”

“This accuracy is satisfactory, as we were able to separate the peak maxima of identical lipid molecular species differing by a single ω -position (e.g. n-9 and n-10) on average by ten (30-minute gradient) and eighteen seconds (60-minute gradient), respectively.”

These RT differences are experimentally verified by isotope-labeled standards. The RT differences can also be seen in our extensive RT-DB (Supplementary Table 1).

Figure 5 presents a case where the two FA chain isomers PC 18:0/22:5(n-3) and PC 16:0/24:5(n-6) coelute. Our approach is able to correctly assign the ω -positions of both isomers as it integrates both, MS/MS information for FA chain verification, and RT information derived from experimentally verified stable isotope labeled species. In contrast, the multiplexed EAD spectrum does not allow for an unambiguous identification.

- 2) *(Figure 6) The chromatogram shows three peaks; however, with RPLC, many phospholipid species with similar fatty acyl/alkyl chain lengths are expected to co-elute in this retention time window. Thus, assigning these three peaks to the n-6, n-7, and n-9 isomers of PI 18:0/20:3 is unconvincing.*

Supplementary Figure 8, which was present already in the initial submission, clearly demonstrates that at the isolated m/z window (DDA), no phospholipid species with other ‘fatty acyl/alkyl chain lengths’ coelute. Also, the lower m/z regions of the spectra of Supplementary Fig. 9 (not shown to provide a clearer view on the C=C location fragments) also reveal that only PI 18:0/20:3 elutes at these retention times (see **Figure R1 below**):

Figure R1: Full spectra of the zoomed spectra in Supplementary Fig. 9 b-d. **a** Spectrum acquired at the RT of PI 18:0/20:3(n-6). **b** Spectrum acquired at the RT of PI 18:0/20:3(n-7). **c** Spectrum acquired at the RT of PI 18:0/20:3(n-9). These spectra reveal that PI 18:0/20:3 are the only species eluting at the retention times of the corresponding ω -position isomers, which are isolated at an m/z of 906.607 for MS/MS fragmentation. Fragments indicative for the lipid class are depicted in orange color, and fragments indicative for FA 18:0 and FA 20:3 in red and blue color, respectively.

3) The authors present EAD-MS analysis of these peaks in Supplementary Fig. 9. However, most of the peaks are barely distinguishable from background noise, making the identification of C=C locations unreliable. Furthermore, co-eluting phospholipids at this retention time are likely to generate overlapping signals, which could lead to mischaracterization of the lipid species

The LC=CL extension pertains to the identification of the ω -positions. The fragments revealing the ω -position (fragments n6, n7 and n9 in Supplementary Fig.9 a, b, and c, respectively) are 7-12 times above the noise level, which is typically sufficient for an identification.

Reviewer #3

- 1) *The authors did an excellent job in addressing all my comments adequately. The described methodology is original and represents an important contribution to lipidomics studies down to the double bond positional level. Therefore, to my opinion, publication in Nature Communications is justified after clarification the following minor issue:*

In the main manuscript, page 9, line 219ff it is stated: "In the provided example, PC 18:0/22:5(n-3) coelutes with PC 18:0/24:5(n-6). Such coeluting isomers are readily identified by the LC=CL with high confidence, as the algorithm assigns ω -positions only when MS/MS evidence is available for lipid molecular species." Should it not be the isomer PC 16:0/24:5(n-6) instead?

And in line 222ff "As this example illustrates, knowledge of the sum composition alone, i.e. PC 38:5, does not suffice for an assignment of C=C positions based on RT." Should this not be "PC 40:5" for consistency?

We very much appreciate that Reviewer #3 recognizes that the "described methodology is original and represents an important contribution to lipidomics studies down to the double bond positional level. Therefore, to my opinion, publication in Nature Communications is justified".

Thank you very much for spotting the two typos. We have corrected them.